# MRC-5 Human Lung Fibroblasts Alleviate the Genotoxic Effect of Fe-N Co-Doped Titanium Dioxide Nanoparticles through an OGG1/2-Dependent Reparatory Mechanism

**DOI:** 10.3390/ijms24076401

**Published:** 2023-03-29

**Authors:** Bogdan Andrei Miu, Ionela Cristina Voinea, Lucian Diamandescu, Anca Dinischiotu

**Affiliations:** 1Department of Biochemistry and Molecular Biology, Faculty of Biology, University of Bucharest, 91-95 Splaiul Independentei, 050095 Bucharest, Romania; miu.bogdan-andrei@s.bio.unibuc.ro (B.A.M.); anca.dinischiotu@bio.unibuc.ro (A.D.); 2Research Institute of the University of Bucharest—ICUB, University of Bucharest, 050657 Bucharest, Romania; 3National Institute of Materials Physics (NIMP), Atomistilor 405A, 077125 Magurele, Romania; diamand@infim.ro

**Keywords:** titanium dioxide, P25 Degussa nanoparticles, iron doping, nanoparticle genotoxicity, human pulmonary fibroblasts, MRC-5 cells, OGG1/2 DNA repair glycosylase, base excision repair

## Abstract

The current study was focused on the potential of pure P25 TiO_2_ nanoparticles (NPs) and Fe(1%)-N co-doped P25 TiO_2_ NPs to induce cyto- and genotoxic effects in MRC-5 human pulmonary fibroblasts. The oxidative lesions of P25 NPs were reflected in the amount of 8-hydroxydeoxyguanosine accumulated in DNA and the lysosomal damage produced, but iron-doping partially suppressed these effects. However, neither P25 nor Fe(1%)-N co-doped P25 NPs had such a serious effect of inducing DNA fragmentation or activating apoptosis signaling. Moreover, oxo-guanine glycosylase 1/2, a key enzyme of the base excision repair mechanism, was overexpressed in response to the oxidative DNA deterioration induced by P25 and P25-Fe(1%)-N NPs.

## 1. Introduction

In recent years, NPs have become a major development opportunity for biomedical [1], agricultural [2], and different industrial applications such as electronics [3], aerospace and automotive coatings [4], active food packaging [5], or environmental remediation [6]. According to some estimates, the most produced metal-based nanomaterials worldwide are made of silver, titanium, zinc, or gold [7,8]. In particular, TiO_2_ NPs are widely used in commercially available products or pilot applications, including cosmetics and sunscreens, paints, food products and active packaging, photoactive cement, or innovative textiles [9,10,11,12], due to their special optical and photocatalytic characteristics. 

Different studies proved that TiO_2_ NPs, especially anatase-rutile mixtures, exhibit cyto- and genotoxicity [13,14,15], which probably derive from their ability to generate excessive ROS levels that could determine oxidative stress when the cellular antioxidant systems are overwhelmed. The toxicological response of living systems to TiO_2_ NPs depends on the particles’ physicochemical properties that can be modulated by various procedures [16,17]. TiO_2_ is present in several stoichiometric (anatase, rutile, brookite) [18,19] and non-stoichiometric crystalline structures [20,21], the toxicity of the first ones being exhaustively investigated over time. Anatase and rutile are the main crystalline forms of TiO_2_. Evidence shows that their toxic effects might differ when nano-dimensioned particles under ultraviolet or visible irradiation are considered [22,23]. Concerning the non-stoichiometric forms, results on their toxicological effects are nearly inexistent. To the best of our knowledge, only one study investigated the biological activity of TiO_2−x_, reporting eryptosis via ROS and Ca^2+^ signaling [24].

Within the nano-size range, smaller NPs are generally considered more cytotoxic because of their large specific surface area, which probably leads to the formation of higher amounts of ROS [25,26,27]. Agglomeration of small TiO_2_ NPs might enhance toxicity, while larger agglomerated NPs could lessen that effect [28]. The health risks of NPs are currently disputed, probably because of the heterogenic characteristics of the investigated NPs. However, oxidative stress is generally accepted as their main source of toxicity [29,30]. 

Nowadays, oxidative stress can be considered a serious threat to human health, as it is associated with the development of different metabolic chronic disorders [31,32], neurodegenerative diseases [33], or carcinogenesis [34]. Different external factors, such as UV radiation [35], an unbalanced diet [36], drugs [37,38], pesticides [39], or air pollution [40], can contribute to the excess generation of free radicals, overwhelming the enzymatic antioxidant defense mechanism of cells. 

The oxidative stress might become dangerous at the cellular level when not sufficiently managed. Its negative effects emerge from the damage of biomolecules caused by excess reactive oxygen species (ROS). These affect the integrity and potential of cellular membranes, including endomembranes, by lipid peroxidation [41,42]. Moreover, ROS can impair nucleotides, leading to mutations [43] or DNA lesions [44].

The uptake of TiO_2_ NPs into the body tissues can occur through inhalation, ingestion, or dermal contact. Recently, the stakeholders of the European Commission agreed, based on the opinion of the European Food Safety Authority, to restrict the use of TiO_2_ as a food additive [45], hence preventing gastrointestinal exposure. While cutaneous application of TiO_2_ NPs-based sunscreens is generally considered safe [46,47], inhalation remains the main route of exposure to nanoscale particles, as they easily diffuse into the atmosphere. The International Agency for Research on Cancer raised concerns regarding the safety of nanoscale TiO_2_ more than 10 years ago, hence its classification in a group of substances that possibly trigger cancer through inhalation [48]. In addition, different regulations concerning cosmetic products that might lead to the inhalation of TiO_2_ NPs were introduced as time passed [49,50]. Different countries also established maximum occupational exposure levels to TiO_2_ NPs [51,52]. 

Genotoxicity is usually accepted as an indicator of NP-induced carcinogenesis. Different studies showed that TiO_2_ NPs are mainly distributed through vesicular structures and are up-taken without harming the cellular organelles [53,54,55]. In contrast, some authors claimed that NPs could interact with different organelles, including the nucleus [56,57,58]. However, experimental results suggest that the direct interaction of DNA molecules with TiO_2_ NPs is improbable. Only small TiO_2_ NPs (dimensions) could be observed in the nucleus occasionally [59,60], and therefore their amount was probably insignificant compared to the proportion of particles that were accumulated in the cytoplasm. Accordingly, diffusion of ROS into the nucleus is possibly the main mechanism causing TiO_2_-induced genotoxicity in the form of oxidized bases, micronuclei, and double-stranded breaks [61,62].

Considering the abovementioned regulations and the scientific data that support the negative effects of TiO_2_ NPs on health, researchers were forced to design strategies to attenuate the toxicity thereof. Different molecules coupled to the surface of particles during or after their synthesis might be effective in reducing ROS production. Antioxidants appear to be a straightforward solution [63], but also doping with metallic ions might be effective. Ghiazza et al. [64] found that doping TiO_2_ NPs with iron could alleviate their ability to induce oxidative stress in human keratinocytes. Co-doping with iron ions could provide a supplementary advantage to TiO_2_ NPs, as it narrows their energy band gap [65]. Therefore, less energy is required to trigger photocatalysis, a physical property exploited in some applications [66]. Normally, the photocatalytic effect of TiO_2_ NPs manifests stronger in UV light. On the contrary, scientific evidence indicated that iron doping could render nanoscale TiO_2_ become photoexcited by exposure to low visible radiation [67]. It is worth mentioning that ion doping does not always act as an inhibitor of ROS [68,69], and therefore its effect might be dependent on NPs’ characteristics, including the ion nature and concentration.

Analyzing the above information, it can be noted that the effect of iron doping on the toxicity of TiO_2_ NPs might be beneficial for the future of nanotechnology and needs to be further explored. Therefore, our study aimed to compare the cytotoxic and genotoxic response of MRC-5 human pulmonary fibroblasts exposed to TiO_2_ P25 NPs and Fe(1%)-N doped TiO_2_ P25 ones, respectively. Also, lysosomal modifications and DNA integrity were investigated in relation to the oxidative lesions induced by the tested NPs. 

## 2. Results

### 2.1. Physicochemical Characteristics of TiO_2_ NPs

The two types of TiO_2_ NPs that were used in the present work were (i) commercially available P25 NPs and (ii) the same NPs co-doped with Fe and N atoms by direct impregnation in an aqueous solution of 1% FeCl_3_ and in the presence of urea (P25-Fe(1%)-N) (see Section 4.1.). The concentration of FeCl_3_ that we chose was mainly based on a previous paper from our group where P25 NPs impregnated by dispersion in 1% FeCl_3_ had an enhanced photocatalytic effect in both long-wave UV (368 nm) and visible light (610 nm) [70]. Moreover, Kalantari et al. showed that co-doping with Fe and N considerably increased the TiO_2_ NPs’ photocatalytic activity compared to mono-doped TiO_2_ NPs [71].

There is evidence that a higher amount of Fe atoms on TiO_2_ NPs’ surface would improve the photocatalytic effect of NPs [72,73,74]. However, it was already shown that the phototoxicity of TiO_2_ NPs could be proportionally increased by 1% to 10% Fe-doping due to the generation of oxidizing agents via the Fenton reaction [74]. Therefore, we considered a low amount of dopant would be more appropriate for investigating the potentially toxic effects of pulmonary exposure to Fe-N doped TiO_2_ NPs.

The chemical content of P25 NPs and P25-Fe(1%)-N NPs was revealed by X-ray photoelectron spectroscopy (XPS) measurements (Figure 1a) and the corresponding binding energies (Table 1). The results proved the presence of Ti and O atoms in both types of NPs (Figure 1b,c). Also, Fe and N atoms were identified in the P25-Fe(1%)-N sample (Figure 1d,e). Moreover, P25 NPs were made of stoichiometrically structured TiO_2,_ as revealed by the ratio of 2.02 between the main signals of Ti 2p_3/2_ (458.65 eV) and O 1s (529.98 eV). The O 1s peaks near 532 eV might be assigned to hydroxyl groups or adsorbed water molecules on the surface, and the 530–531 eV peaks to Ti-O chemical bonds, respectively. The signal at 710.40 eV is characteristic of Fe 2p_3/2_, revealing that P25-Fe(1%)-N NPs contained Fe^3+^. The peak at 399.62 eV might be assigned to oxidized nitrogen, i.e., O-Ti-N bindings. Also, the N 1s peak at 401.19 eV usually reflects interstitial nitrogen. The ratios between intensities of the XPS peaks (Fe/TiO and N/TiO) showed that P25 NPs prepared in FeCl_3_ had 2.1% Fe atoms and 0.5% N atoms on their surface.

### 2.2. Oxidative DNA Damage Induced by TiO_2_ NPs in MRC-5 Cells

The concentrations of NPs, i.e., 10 µg/mL and 50 µg/mL, respectively, used by us were chosen based on our previous work [75] in which we proved that P25 NPs could cause a significant increase of oxidative stress in MRC-5 cells in a time- and dose-dependent manner while P25-Fe(1%)-N NPs had no influence on ROS level compared to the control group of cells. 

ROS can damage the cell considerably by impairing the constitutive molecules of cellular structures. One of the damages induced by a high level of ROS is the oxidation of guanosine, a modification that might affect the integrity of DNA molecules. We investigated the impact of TiO_2_ NPs on the DNA molecules of MRC-5 cells by measuring the level of 8-hydroxydeoxyguanosine (8-OHdG), a commonly used marker for DNA oxidative lesions. Our results showed that exposure to P25-Fe(1%)-N NPs could increase the level of 8-OHdG in a time-dependent manner (Figure 2) in MRC-5 cells. The levels of 8-OHdG induced by both doses of P25-Fe(1%)-N NPs and the dose of 10 µg/mL non-doped P25 NPs were generally similar and have not exceeded 130% compared to control after 72 h of exposure. However, the higher dose of P25 NPs caused an increase of the level of 8-OHdG up to 235% compared to the control after MRC-5 cells were exposed for 24 h. Also, 8-OHdG concentration was lowered in the cells as time passed, reaching 166% compared to the control at 72 h. Interestingly, the reduction of 8-OHdG recorded at 72 h at 50 µg/mL P25 NPs contrasts with the high level of ROS measured previously by us in MRC-5 cells [75].

### 2.3. Influence of TiO_2_ NPs on the Morphology of MRC-5 Cells

Actin cytoskeleton plays a key role in the mechanical support of cells, also defining their morphology. Fluorescent microscopy images displayed in Figure 3 showed that TiO_2_ NPs had no negative impact on the MRC-5 cells’ actin cytoskeleton organization. The microscopic images suggested that MRC-5 cells maintained their fibroblast-like morphology regardless of the conditions applied in our study (type of TiO_2_ NPs, concentration of NPs, exposure time). Normally, these lung fibroblasts are elongated spindle-shaped bipolar cells. No disrupted filaments or cytoskeleton rearrangements were observed, while bundles of F-actin appeared very dense, indicating a high cellular density.

### 2.4. Influence of TiO_2_ NPs on Lysosomes’ Formation and Lysosomal Membrane Integrity in MRC-5 Cells

There are no statistically significant differences regarding the accumulation of lysosomes inside MRC-5 cells exposed to P25 and P25-Fe(1%)-N NPs (Figure 4a,b). However, we noted that the lysosome quantity increased by ∼14–18% compared to the control when the doses of 50 μg/mL at 72 h were applied. The distribution of cathepsin B suggested that the membrane of lysosomes was affected by the 72 h exposure to TiO_2_ NPs. Cathepsin B is a key proteolytic enzyme localized in lysosomes under physiological conditions. Therefore, when labeled with Alexa Fluor 594, cathepsin B is present in fluorescent red vesicles in healthy cells, as can be observed in our control cells (Figure 4c). 

When MRC-5 cells were treated with P25 and P25-Fe(1%)-N NPs, the red signal appeared in a diffused pattern, indicating that the lysosomal membrane was permeabilized and cathepsin B was released into the cytosol. Even though permeabilization occurred in all treated pulmonary fibroblasts, images showed that the red signal is more clustered in cells exposed at P25-Fe(1%)-N NPs, suggesting their effect on lysosome integrity is less pronounced than the one of P25 NPs.

### 2.5. Effect of TiO_2_ NPs on the Integrity of DNA from MRC-5 Cells

Considering the generation of ROS and oxidative lesions induced by TiO_2_ NPs in MRC-5 cells, we further decided to investigate whether they critically affect the integrity of DNA molecules. Fragmentation of DNA was investigated by Comet assay that indicated no significant changes between the different conditions tested, although raised levels of DNA oxidation might be considered a marker of double-strand breaks. It can be visually observed that no small fragments of DNA detached and migrated faster, as in the case of the positive control (Figure 5a). The damage of DNA molecules was expressed in percentages of DNA in the comet tail. Based on the quantified fluorescence (Figure 5b), the tail DNA% in samples varied in the 2.4–4.6% range, while in the negative control cells, it has not exceeded 4%. These results could strengthen the evidence that a molecular mechanism ameliorates the TiO_2_ NPs-dependent oxidative damage of DNA within MRC-5 fibroblasts.

### 2.6. Cell Death Signaling in MRC-5 Cells Exposed to TiO_2_ NPs

To investigate whether the oxidative lesions produced by TiO_2_ NPs generated damages that trigger cell death signaling, protein expression of cathepsin B, p53, caspase-8, -9, and -3 were quantified using Western Blot analyses. Cathepsin B presented relatively constant levels in MRC-5 cells exposed to P25 and P25-Fe(1%)-N NPs regardless of the exposure time or dose applied (Figure 6a,b). 

On the contrary, TiO_2_ NPs significantly changed the expression of p53 protein in MRC-5 cells in a time- and dose-dependent manner. After 24 h of exposure to 10 μg/mL of P25 and P25-Fe(1%)-N NPs, the level of p53 decreased by 3%, respectively, 12% relative to the control. The expression of p53 started to drop considerably when the highest dose of TiO_2_ NPs was applied. At 24 h of exposure, P25 NPs led to the diminution of p53 expression by 64%. By comparison, the effect of iron-doped TiO_2_ NPs was slightly milder, leading to a decreased expression by nearly 38% relative to the control. However, the results indicated that the inhibitory effect of TiO_2_ NPs on p53 expression was more evident as time went on. Thus, p53 expression in NP-treated MRC-5 cells exhibited a massive reduction regardless of the dose or exposure time. As can be seen in Figure 6c,d, the level of expression of p53 dropped below 10% relative to the control and was totally suppressed by the treatment with 10 μg/mL and 50 μg/mL of P25 NPs, respectively. 

Both p53 and cathepsin B are involved in the initiation of programmed cell death pathways [76,77]. However, in contrast with our results, p53 normally undergoes overexpression during apoptosis [78]. In the present study, we proved that neither initiator caspases-8 and -9 nor the effector caspase-3 were activated by the TiO_2_ NPs applied to pulmonary fibroblasts. Based on the molecular mass, protein bands displayed on the obtained blot profiles corresponded to the uncleaved, i.e., non-activated procaspases (Figure 6e). The measured level of the apoptosis-inducing markers, i.e., cathepsin B, p53, caspase-8, -9, and -3, correlated well with the high DNA integrity revealed by the Comet assay.

### 2.7. The Reparatory Role of 8-oxoguanine DNA Glycosylase in MRC-5 Cells Exposed to TiO_2_ NPs

As the results suggested so far, a molecular reparatory mechanism might be the reason for the low genotoxic effect of TiO_2_ NPs on MRC-5 cells. Therefore, we decided to investigate a key enzyme involved in the base excision repair, which can recognize oxidized guanine within the DNA, namely OGG1/2. The protein expression of OGG1/2 was assessed by Western Blot analysis, with the representative blot profiles displayed in Figure 7a. Interestingly, our results showed that the level of OGG1/2 slightly decreased when MRC-5 cells were exposed for 24 h at both doses of P25 or P25-Fe(1%)-N NPs. The expression exhibited an insignificant diminution of at most 6% relative to the control. The effect of TiO_2_ NPs on the level of OGG1/2 became evident at 72 h of exposure when the reparatory protein exhibited a substantial overexpression (Figure 7b). In general, P25 and P25-Fe(1%)-N NPs caused the doubling of OGG1/2 expression level, confirming that the innate base excision repair mechanism coped with the oxidative damage induced by TiO_2_ NPs and thus maintained the integrity of DNA molecules. The level of OGG1/2 was slightly higher in the pulmonary fibroblasts treated with P25-Fe(1%)-N NPs in comparison with the one measured in cells exposed to undoped TiO_2_ NPs. Differences in OGG1/2 expression were comprised between 83.5 and 125% relative to the control.

## 3. Discussion

Considering iron doping might intensify the use of TiO_2_ NPs in consumer goods, we chose to investigate the possible associated toxicological risks due to the tuning of their photocatalytic properties toward visible illuminance. Therefore, we compared the toxicity of P25 TiO_2_ NPs doped with iron and nitrogen with the same undoped NPs. The doses used by us were based on our previously published work [75] as well as on representative papers [79,80,81,82]. 

Moreover, inhalation is a major route by which TiO_2_ NPs enter the human body; therefore, we have chosen MRC-5 cells, which are human pulmonary fibroblasts, as the experimental model.

Previous studies stated that TiO_2_ NPs could also interact with the microtubules and other components of the cellular cytoskeleton [83,84]. We decided to investigate the influence of P25 and P25-Fe(1%)-N NPs on the actin cytoskeleton as they could provide valuable information regarding the morphology of treated cells. Even though some reports show that TiO_2_ NPs can disrupt actin filaments [85,86], we did not observe any significant changes in the organization of the cytoskeleton between exposed samples and control cells. In addition, considering the role of the actin cytoskeleton in internalization mechanisms [87], endocytosis of the P25 and P25-Fe(1%)-N NPs with sizes about 50 nm might be produced to a lesser extent in the MRC-5 cells. This fact is supported by the dimension of the large aggregates of TiO_2_ NPs formed [88], which would not be able to enter the cells through caveolae (with a diameter between 50nm and 80 nm) or clathrin-mediated endocytosis (with a diameter of ≈120 nm) [89]. However, Thurn et al. [90] stated that the uptake of aggregates could be possible through macropinosomes with a dimension of 500–2000 nm. 

In our previous work [75], we already showed the significant difference between the ability of P25 and P25-Fe(1%)-N NPs to produce ROS. While P25 NPs could induce high levels of ROS in a time and dose-dependent manner, doping with iron ions totally suppressed the generation of oxidative stress [75]. Similarly, iron doping inhibited the production of TiO_2_-induced ROS in HaCaT keratinocytes [64]. In contrast, doping TiO_2_ NPs with copper led to higher ROS production in A549 cells [69], and doping them with zinc enhanced the oxidative stress induced in MCF-7 cells [68]. 

Moreover, our previous paper [75] investigated the effect of TiO_2_ NPs on the enzymatic antioxidant mechanism of MRC-5 cells. When the ROS level exceeded the neutralizing ability of antioxidant enzymes, some of the free radicals began to impair intracellular biomolecules. A part of the oxygen-derived free radicals produced lipid peroxidation that attacks organelles’ membranes, while others damage DNA after entering the nucleus as well as proteins [91]. 

Some of the most commonly studied biomarkers indicating oxidative damage on DNA molecules, are 8-hydroxylated guanine species, mainly 8-oxoguanine (8-oxoG) and its isomer, 8-OHdG. In our study, P25 NPs increased the level of 8-OHdG in MRC-5 cells in a time-dependent manner, the results being in accordance with the level of ROS produced. Interestingly, we observed an attenuated but significant increase of 8-OHdG level in pulmonary fibroblasts treated with P25-Fe(1%)-N NPs. This might be explained by the fact that iron-doped TiO_2_ NPs could generate some reactive species in the first hours of exposure that had probably produced their effects before initiating the antioxidant mechanisms [75]. We considered this might represent preliminary evidence that human pulmonary fibroblasts are able to counteract excessive oxidation caused by TiO_2_ NPs.

Some studies showed that the increased level of guanine oxidation products within cells might be linked in certain circumstances with DNA fragmentation [92,93]. On the contrary, our results indicated that the integrity of DNA molecules from MRC-5 cells was not affected by the increased level of 8-OHdG caused by exposure to TiO_2_ NPs. Similarly, Hackenberg et al. [94] showed that TiO_2_ NPs did not induce DNA fragmentation in lymphocytes obtained from the peripheral blood of human donors. Bhattacharya et al. [95] obtained the same result when they applied TiO_2_ NPs on both BEAS-2B (normal human bronchial epithelial cells) and IMR-90 (normal human pulmonary fibroblasts) cell cultures, showing that IMR-90 cells exhibited high levels of 8-OHdG after 24 h of exposure to TiO_2_ NPs. Contrariwise, the potential of TiO_2_ NPs to induce DNA double-stranded breaks was demonstrated in HUVEC cells. The genotoxic effect of TiO_2_ NPs was more pronounced as their particle size diminished, producing more DNA damage [96]. 

Besides damages caused by TiO_2_ NPs-induced oxidative stress on DNA molecules, we investigated the influence of this on the membrane of lysosomes from MRC-5 cells. So far, different studies have demonstrated that NPs could induce the permeabilization of lysosomal membranes. For example, Li et al. [97] found that the membrane of lysosomes from MRC-5 cells could be affected by Au NPs. In addition, membrane permeabilization was induced in THP-1 cells by ZnO NPs [98], in HepG2 NPs by Ag NPs [99], or in 3T3 cells by Si NPs [100]. The previously mentioned studies that investigated Au and Si NPs associated the damages caused on lysosomal membranes with an increase in the generation of ROS. 

One of the roles of lysosomes is to enzymatically digest spent cellular organelles. As high levels of ROS in MRC-5 cells treated with TiO_2_ NPs might damage different intracellular structures, an increase in the number of lysosomes was expected. Our results suggested that the number of lysosomes was not significantly influenced by P25 and P25-Fe(1%)-N NPs exposure. However, the lysosomal membrane was significantly impaired at 72 h of exposure. The damaged membrane of lysosomes probably allowed the release of lysosomal content, especially cathepsins, enzymes that can be involved in activating caspase-dependent cell death pathways [101]. Neither the expression of the p53 protein nor those of caspase-3, -8, and -9 indicated that apoptosis was activated in MRC-5 cells by TiO_2_ NPs, although we observed that cathepsin B diffuses from lysosomes into the cytosol. The insignificant differences between the expression of cathepsin B validated that the diffuse red signal obtained through immunofluorescence resulted only from the lysosomal membrane permeabilization. As cathepsin B is a lysosome-resident protein, the result confirmed that TiO2 NPs did not significantly influence lysosomal formation in MRC-5 cells.

Besides the innate antioxidant defense system that acts directly on generated ROS, eukaryotic cells can cope with oxidative damage of DNA due to different reparatory mechanisms, including the base excision repair (BER) mechanism. OGG1/2 has a crucial role in the removal of oxidized guanine species, being the enzyme responsible for their recognition, hence the initiation of the BER process [102]. We found that MRC-5 cells overexpressed OGG1/2 when exposed to both P25 and P25-Fe(1%)-N TiO_2_ NPs for 72 h, suggesting the BER mechanism was induced. The constant level of OGG1/2 noticed after 24 h of exposure might be explained by a delay between transcription and translation processes. The first result that suggested a reparatory mechanism had been activated was the decrease of 8-OHdG level in the case of 50 μg/mL P25 NPs exposure at 72 h and further the unaffected DNA integrity revealed by Comet assay.

Du et al. [103] revealed that OGG1 is overexpressed in a dose-dependent manner in human hepatocytes L02 by a combined treatment of TiO_2_ NPs and lead, whereas Zijno et al. [104] showed that OGG1 level increased in human colon Caco-2 cells following treatment with TiO_2_ NPs. Also, Xia et al. [105] found that human kidney HEK293T cells express OGG1 in response to the oxidative damage caused by TiO_2_ NPs that act synergistically with CdCl_2_.

In contrast with our results, control of BER activity is managed by p53 through its ability to regulate the cell cycle [106]. We found that the expression of p53 was totally inhibited. Therefore, the point mutations caused by 8-OHdG in the sequence of DNA [43] might have been transmitted during cell division prior to the activation of the reparatory mechanism. However, the BER pathway can function in a p53-independent manner, as other proteins might arrest the cell cycle [107]. 

## 4. Materials and Methods

### 4.1. Physicochemical Characterization of TiO_2_ NPs

Two types of TiO_2_ NPs were used in this study: (i) Degussa P25 (Aeroxide^®^ P25) purchased from Sigma Aldrich (St. Louis, MO, USA) and (ii) Degussa P25 co-doped with Fe and N atoms that were obtained experimentally by direct impregnation in 1% FeCl_3_ 6H_2_O and urea. The method of impregnation of TiO_2_ NPs with Fe and N, as well as the characteristics of the two types of TiO_2_ NPs, were described in detail in the previous publications of our research group [70,88]. Briefly, powders of P25 and P25-Fe(1%)-N NPs consisted of approx. 83% anatase (with a crystallite size of around 30 nm) and approximately 17% rutile (with a crystallite size of around 50 nm) [70]. Moreover, our group showed that these types of TiO_2_ NPs formed large aggregates when they were suspended in MEM supplemented with 10% FBS. Zeta potential values around –10 mV also confirmed the low stability of TiO_2_ NPs [88].

XPS measurements provided in this work were obtained in an analysis chamber using a monochromatized Al K_α1_ X-ray source (1486.74 eV). The electrons were analyzed with a 150 mm hemispherical electron energy analyzer (Phoibos, Specs Gmbh, Berlin, Germany).

### 4.2. Cell Culture and Treatment with TiO_2_ NPs 

MRC-5 human lung fibroblasts purchased from American Type Culture Collection (ATCC, catalog no. CCL-171) were cultured in vitro in Eagles minimum essential medium (MEM; Gibco/Invitrogen, Carlsbad, CA, USA) at 37 °C and in a humified atmosphere with 5% CO_2_. MEM containing 2 mM L-glutamine, 0.1 mM sodium pyruvate, and 4.5 g/L glucose was supplemented with 10% fetal bovine serum (FBS; Gibco/Invitrogen, Carlsbad, CA, USA). Replacement of the growth medium with a fresh one was done every two days. Sub-cultivations were performed when cells reached ~80% confluence. For sub-cultivation, MRC-5 cells were detached using a solution of 0.25% (*w*/*v*) Trypsin with 0.53 mM EDTA (Sigma Aldrich, St. Louis, MO, USA) and split into other culture flasks. 

In this experiment, MRC-5 human lung fibroblasts were exposed to 10 and 50 μg/mL TiO_2_ NPs for 24 and 72 h. Stock suspensions of 2 mg/mL TiO_2_ NPs (P25 and P25-Fe(1%)-N) were prepared by adding 10 mg of each NP’s type in 5 mL of phosphate-buffered saline (PBS), pH ≈ 7.4. For improving particles’ dispersion, suspensions were sonicated for 5 min at room temperature using the ultrasonic processor UP50H (Hielscher Ultrasonics GmbH, Teltow, Germany). Then, stock suspensions were exposed for 30 min to UV light to be sterile when used. MRC-5 cells were detached as described above and seeded into 75 cm^2^ culture flasks. P25 and P25-Fe(1%)-N NPs were added directly into the culture medium at the abovementioned final concentrations. Cells used as the control for each assay underwent the same procedures but were grown in an NP-free culture medium.

### 4.3. Measurement of 8-Hydroxy-2′-Deoxyguanosine Level

The level of 8-OHdG was measured using a commercially available enzyme-linked immunosorbent assay (ELISA) kit purchased from Abcam (ab201734; Cambridge, UK). Previously, DNA from MRC-5 cells exposed to TiO_2_ NPs was isolated and quantified. Afterward, the DNA was digested with P1 nuclease and treated with alkaline phosphatase; thus, nucleotides were transformed into nucleosides. Further, DNA samples were processed using the 8-OHdG ELISA kit according to the manufacturer’s instructions, and finally, their absorbance was measured at 450 nm using a microplate reader (TECAN GENios, Grödig, Austria).

### 4.4. Fluorescence Microscopy Analysis

Fluorescent staining was used to analyze the actin cytoskeleton morphology and dynamic, lysosomes’ number and density, as well as cathepsin B localization. To observe actin filaments, MRC-5 cells cultured in flasks and exposed to TiO_2_ NPs were fixed with 4% paraformaldehyde for 20 min at room temperature. Then, cell membranes were permeabilized with a mixture of 0.1% Triton X-100 and 2% bovine serum albumin (BSA) for 30 min. F-actin was labeled by incubating the cells for 1 h with 20 μg/mL phalloidin-fluorescein isothiocyanate (FITC; Sigma Aldrich, St. Louis, MO, USA). The staining of cell nuclei has been done by 4′,6-diamidino-2-fenilindol (DAPI; Molecular Probes, Life Technologies, Carlsbad, CA, USA). Images of the actin cytoskeleton were acquired using the inverted fluorescence microscope Olympus IX71 (Tokyo, Japan).

The fluorescent staining of lysosomes was performed by incubating MRC-5 cells with 100 nM LysoTracker Green DND-26 (Molecular Probes, Invitrogen) for 30 min at 37 °C in a humidified atmosphere containing 5% CO_2_. Hoechst 33342 (Molecular Probes, Invitrogen) was used to counterstain cell nuclei. Images of stained lysosomes were taken with Olympus IX71 inverted fluorescence microscope (Tokyo, Japan). Green fluorescence intensity in different fields of view per each sample was quantified using the ImageJ 1.53u software available online at https://imagej.nih.gov/ij/ (National Institute of Health, Bethesda, MD, USA) and displayed as a mean relative to the control.

Immunofluorescent localization of cathepsin B was performed by seeding MRC-5 cells on coverslips at a density of 2 × 10^4^ cells/cm^2^. After fibroblasts were allowed to adhere overnight, they were exposed to TiO_2_ NPs, as described in Section 4.2. Further, MRC-5 cells underwent fixation and permeabilization as described above in the case of F-actin. Cathepsin B was labeled by incubating cell plates (overnight, 4 °C) with Alexa Fluor 594-coupled anti-cathepsin B antibody (Santa Cruz Biotechnology Inc., Dallas, TX, USA). The staining of cell nuclei has been done by DAPI. Labeled cathepsin B was visualized at 60x objective of the fluorescence microscope Nikon Eclipse E200 (Tokyo, Japan). 

### 4.5. Comet Assay

Comet assay was performed using a single-cell electrophoresis kit (Cell Biolabs, INC, San Diego, CA, USA). After exposure to the two types of TiO_2_ NPs, MRC-5 cells were collected, resuspended in PBS, and diluted until the density of 1 × 10^5^ cells/mL was reached. A volume of 10 μL of each cellular suspension was mixed with 100 μL low melting agarose maintained at 37 °C. Further, a volume of 75 μL from this mixture was stretched uniformly in thin films on a Comet glass slide. The agarose was allowed to jellify by incubating the slides on a horizontal surface in the dark at 4 °C for 15 min. Then, cells embedded in agarose were lysed (using the lysis solution within the kit at 4 °C, 60 min) and further treated with an alkaline solution (4 °C, 30 min). Afterward, the slides were washed with deionized water and subjected for 20 min to low voltage horizontal electrophoresis migration (20V). Subsequently, the slides were washed with 70% ethanol. Finally, DNA molecules from the agarose-embedded cells were stained with the Vista Green fluorescent dye. The negative control was represented by MRC-5 cells cultivated in an NP-free growth medium. The positive control underwent the same procedure, but NP-free cultured cells embedded in agarose were exposed at 70 μM H_2_O_2_ (5 min, 4 °C). Images of the comets were acquired using the fluorescence microscope Olympus IX 71 (Tokyo, Japan). Fluorescence from representative images was quantified using the OpenComet plugin within the ImageJ 1.53u software (National Institute of Health, Bethesda, MD, USA) and displayed as a percentage of tail DNA expressed relative to the negative control.

### 4.6. Western Blot Analysis

Western Blot technique was used to determine the expression level of p53, cathepsin B, caspase-3, -8, -9, and OGG1/2 proteins. In advance, total protein extracts of samples were prepared, and their concentration was measured by the Bradford method. Harvested MRC-5 fibroblasts suspended in PBS were subjected to 3 cycles of 30 s ice-assisted sonication using the ultrasonic processor UP50H (Hielscher, Teltow, Germany) to disrupt the cell membranes. Obtained lysates were centrifuged at 3000× *g*, at 4 °C for 10 min, and then each supernatant containing the total protein extract was individually collected and stored at −80 °C until further use.

Cell lysates containing an equal amount of total protein were subjected to sodium dodecyl sulfate-polyacrylamide gel electrophoresis (SDS-PAGE) (90V, 120 min) and then transferred to a polyvinylidene fluoride membrane (PVDF; Millipore, Billerica, MA, USA) at 350 mA for 95 min within a wet transfer unit (Bio-Rad Laboratories, Hercules, CA, USA). For the detection of proteins, PVDF membranes were processed using the WesternBreeze Chromogenic Kit (Invitrogen, Grand Island, NY, USA). A blocking buffer was applied for 30 min, and then the membranes were incubated overnight with the following primary monoclonal antibodies purchased from Santa Cruz Biotechnology Inc. (Dallas, TX, USA): anti-p53, anti-cathepsin B (sc-365558), anti-β-actin (sc-517582), anti-caspase-8 (sc-5263), anti-caspase-9 (sc-56076), anti-caspase-3 (sc-7148), anti-oxoguanine glycosylase 1/2 (OGG1/2; sc-376935). Excess of the primary antibody was released, and membranes were incubated with an alkaline phosphatase-coupled anti-mouse secondary antibody. Afterward, protein bands were revealed using 5-bromo-4-chloro-3-indolyl phosphate/nitroblue tetrazolium (BCIP/NBT). Blot images were acquired with the ChemiDoc XRS+ system (Bio-Rad Laboratories, Hercules, CA, USA) and processed using Image Lab (version 6.1., Bio-Rad Laboratories, Hercules, CA, USA) software. Protein levels were quantified with GelQuant.NET software (version 1.8.2., available online at BiochemLabSolutions.com), the amount of β-actin from each sample being used to normalize protein expression. The results were expressed as percentages relative to the control cells’ protein expression. 

### 4.7. Protein Concentration

Protein concentration was measured using the Bradford method [108]. Briefly, the optical density of the reaction product between Bradford Reagent (Sigma Aldrich, St. Louis, MO, USA) and total protein extracts was measured at 595 nm using a FlexStation 3 Spectrophotometer. Protein concentrations of all samples were calculated based on a BSA standard curve between 0 and 1.25 mg/mL (0–18.8167 μM). 

### 4.8. Statistical Analysis

The means of three independent experiments were expressed as percentages relative to the control ± standard deviation. Statistical differences between each treatment and the control were evaluated using the Student’s two-tailed t-test. The statistical significance was displayed based on the *p* values as follows: * for *p* < 0.05; ** for *p* < 0.01; *** for *p* < 0.001. All the data were analyzed and visualized using GraphPad Prism software (version 8; GraphPad Software Inc., San Diego, CA, USA). 

## 5. Conclusions

Our results could suggest that the oxidative lesions caused by TiO_2_ NPs in human pulmonary fibroblasts could be partially neutralized by co-doping them with low amounts of nitrogen and iron ions. Moreover, the toxic effects of P25-Fe(1%)-N NPs can be considered attenuated compared to the undoped P25, albeit they were not totally suppressed. The main impairments probably produced by ROS in pulmonary fibroblasts were related to the oxidation of DNA components and lysosomal membrane permeabilization that led to the leakage of lysosomes’ content into the cytoplasm. Additionally, overexpression of OGG1/2 in correlation with the integrity of DNA molecules indicated that probably the BER mechanism successfully managed the intranuclear damages induced by TiO_2_ NPs. Therefore, we hypothesized that MRC-5 cells might be more resilient than other cell types to the effects induced by TiO_2_ NPs. This conclusion could also be supported by the fact that pulmonary cells are usually more exposed to exogenous ROS-producing compounds that enter the lungs by inhalation, and their reparatory mechanisms are probably more active. However, other implications might be involved. The inhibited expression of p53 suggested that the cell cycle of pulmonary fibroblasts was not arrested during reparatory processes, as normally happens, indicating that the DNA errors, which probably occurred, might persist during cell division. In conclusion, our study showed that intracellular mechanisms of pulmonary fibroblasts could be stressed by TiO_2_ NPs even though cell viability was not affected. Moreover, iron doping of TiO_2_ NPs might be considered a suitable strategy to attenuate the effects of TiO_2_ NPs on MRC-5 cells. We consider that this research contributes to the knowledge regarding the interaction of doped P25 NPs with molecular mechanisms of in vitro cultured cells and might be a support for the design of safer and more efficient TiO_2_ NPs. 

## Figures and Tables

**Figure 1 ijms-24-06401-f001:**
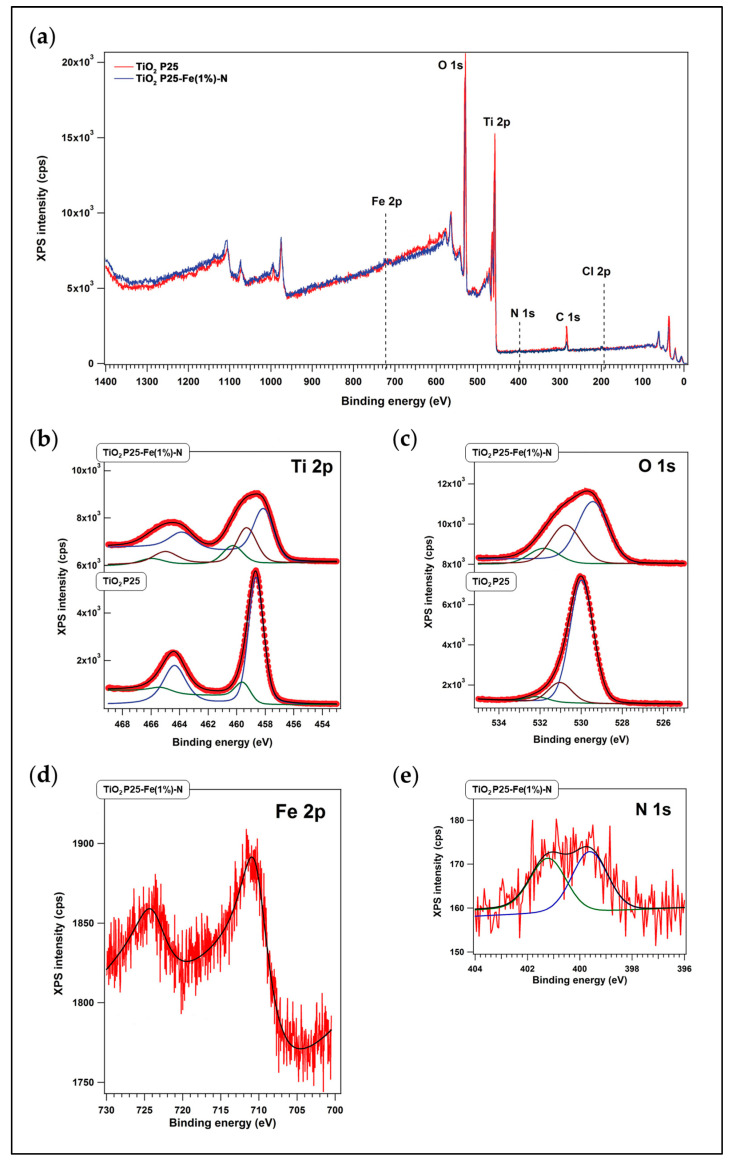
X-ray photoelectron spectroscopy (XPS) spectra of P25 NPs and Fe(1%)-N doped P25 NPs: (**a**) XPS survey spectrum and high-resolution XPS scan spectra over (**b**) Ti 2p, (**c**) O 1s, (**d**) Fe 2p and (**e**) N 1s peaks.

**Figure 2 ijms-24-06401-f002:**
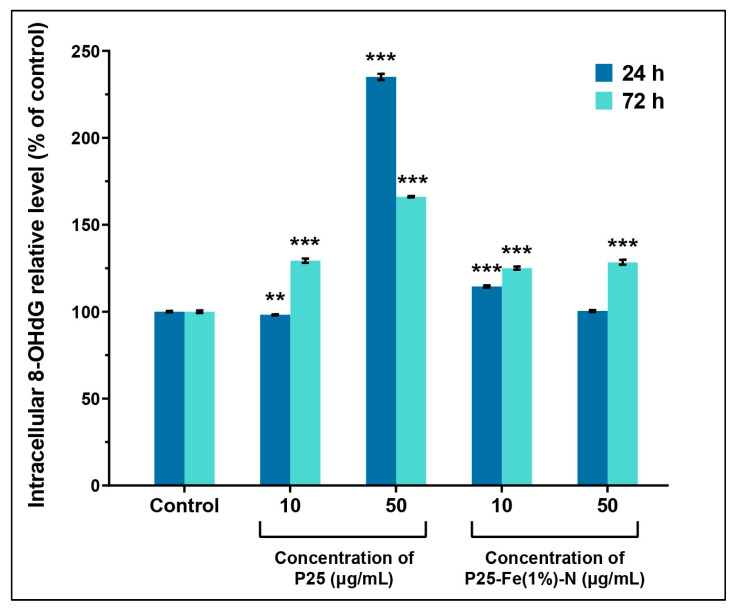
TiO_2_ NPs induced oxidative DNA damage in human lung fibroblasts. 8-OHdG levels in MRC-5 cells treated with different concentrations (10 and 50 μg/mL) of P25 NPs and Fe(1%)-N doped P25 NPs at 24 and 72 h of exposure. Each bar represents the means expressed as % relative to untreated cells ± standard deviation. Statistical significance: ** *p* < 0.01 and *** *p* < 0.001 (comparison of each treatment with the control).

**Figure 3 ijms-24-06401-f003:**
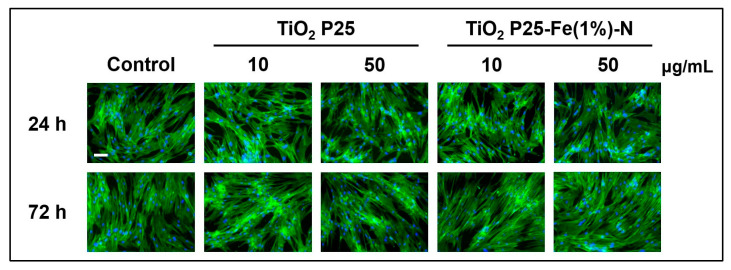
TiO_2_ NPs effect on the morphology of human lung fibroblasts. Fluorescence microscopy images of actin cytoskeleton structure in MRC-5 cells treated with P25 NPs and Fe(1%)-N doped P25 NPs (10 and 50 μg/mL) at 24 and 72 h of exposure. Bundles of F-actin (green) were labeled with phalloidin-fluorescein isothiocyanate (FITC). Nuclei (blue) were stained with DAPI. Scale bar: 100 µm.

**Figure 4 ijms-24-06401-f004:**
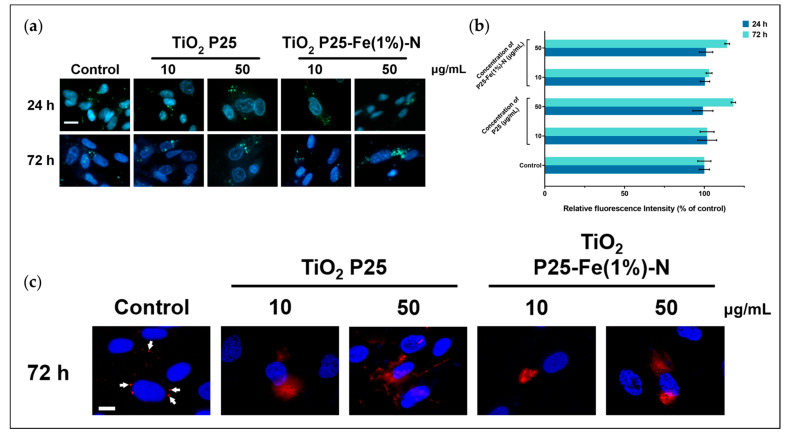
TiO_2_ NPs effect on lysosomes from human lung fibroblasts: (**a**) Representative images of lysosomes (green) labeled with LysoTracker Green and (**b**) quantification of green fluorescence intensity in MRC-5 cells treated with different concentrations (10 and 50 μg/mL) of P25 NPs and Fe(1%)-N doped P25 NPs at 24 and 72 h of exposure. Each bar represents the means expressed as % relative to untreated cells ± standard deviation. Nuclei (blue) were stained with Hoechst 33342. Scale bar: 50 µm; (**c**) Representative images of cathepsin B (red) labeled with Alexa Fluor 594 in MRC-5 cells treated with P25 NPs and Fe(1%)-N doped P25 NPs (10 and 50 μg/mL) at 72 h of exposure. White arrows indicate the vesicular disposition of cathepsin B in control cells. Nuclei (blue) were stained with DAPI. Scale bar: 20 µm.

**Figure 5 ijms-24-06401-f005:**
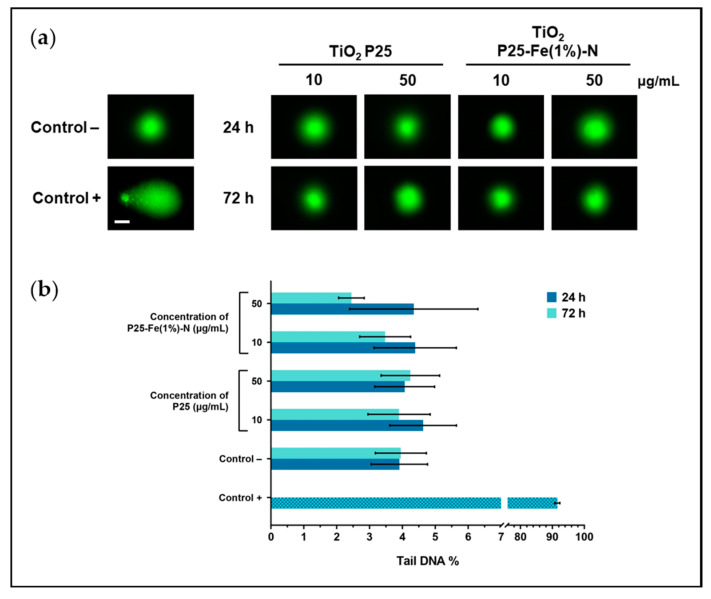
TiO_2_ NPs effect on DNA integrity of MRC-5 cells: (**a**) Representative images obtained by Comet assay (scale bar: 50 µm) and (**b**) quantification of the green fluorescence expressed as tail DNA percent ± standard deviation.

**Figure 6 ijms-24-06401-f006:**
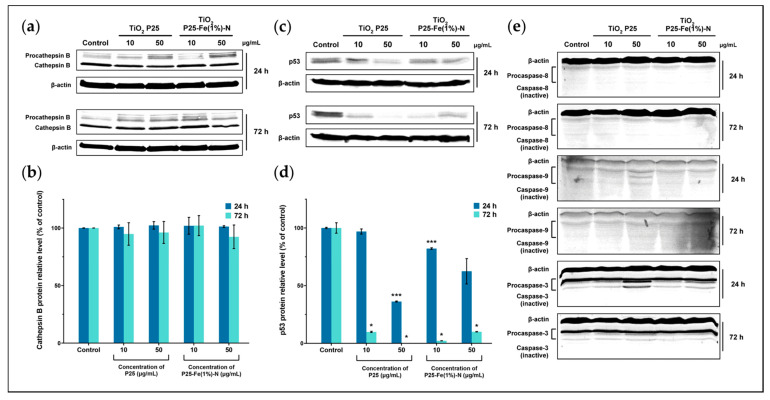
TiO_2_ NPs effect on the expression of: (**a**,**b**) cathepsin B, (**c**,**d**) p53, (**e**) caspase-8, caspase-9, and caspase-3 in MRC-5 cells. Each bar represents the means expressed as % relative to untreated cells ± standard deviation. Statistical significance: * *p* < 0.05 and *** *p* < 0.001 (comparison of each treatment with the control).

**Figure 7 ijms-24-06401-f007:**
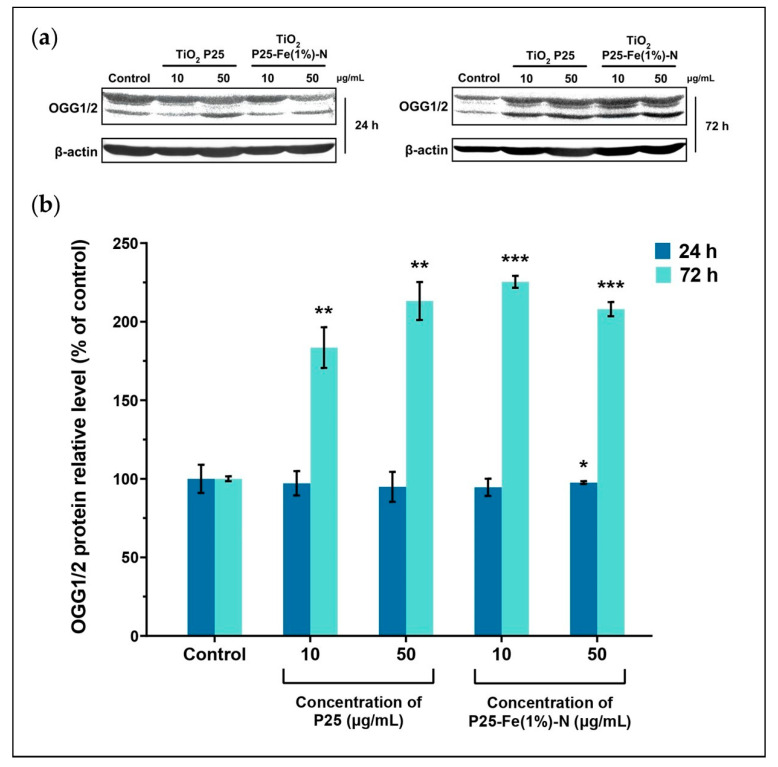
Possible initiation of the base excision repair mechanism through OGG1/2 activation by TiO_2_ NPs: (**a**) Western Blot profile and (**b**) quantification of the expression of OGG1/2 in MRC-5 in response to the treatment with different concentrations (10 and 50 μg/mL) of P25 NPs and Fe(1%)-N doped P25 NPs at 24 and 72 h of exposure. Each bar represents the means expressed as % relative to untreated cells ± standard deviation. Statistical significance: * *p* < 0.05, ** *p* < 0.01, and *** *p* < 0.001 (comparison of each treatment with the control).

**Table 1 ijms-24-06401-t001:** The binding energy values extracted from the deconvolutions of the XPS spectra of P25 NPs and Fe(1%)-N doped P25 NPs.

Sample	Ti 2p_3/2_	O 1s	Fe 2p_3/2_	N 1s
	Binding Energy (eV)			
TiO_2_ P25	458.65 459.63	529.98		
		531.03	-	-
		532.16		
TiO_2_ P25-Fe(1%)-N	458.08	529.42		399.62 401.19
	459.30	530.75	710.40	
	460.27	531.81		

## Data Availability

Data are available on request from the corresponding author.

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
