# Peer review of "MRC-5 Human Lung Fibroblasts Alleviate the Genotoxic Effect of Fe-N Co-Doped Titanium Dioxide Nanoparticles through an OGG1/2-Dependent Reparatory Mechanism"

_ijms, 2023, doi:10.3390/ijms24076401_

Round 1

Reviewer 1 Report

Referee report: “MRC-5 Human Lung Fibroblasts Alleviate the Genotoxic Effect of Fe-N Co-Doped Titanium Dioxide Nanoparticles Through  an OGG1/2-Dependent Reparatory Mechanism

This is a very interesting review article that probably can be recommended for publication, but only after clarifying and detailing some parts of the text.

1.     It is not clear from the second paragraph which structure of the titanium oxide is meant, because TiO2 is known in several crystalline modifications. It is important to note that there are several modification of TiO2 and they give different effects. See, some of them:  Tsebriienko, T.; Popov, A.I. Effect of Poly(Titanium Oxide) on the Viscoelastic and Thermophysical Properties of Interpenetrating Polymer Networks. Crystals 202111, 794. https://doi.org/10.3390/cryst11070794

2.     What effect does the size of nanoparticles have? How important is stoichiometry?

Dorosheva, I.B.; Valeeva, A.A.; Rempel, A.A.; Trestsova, M.A.; Utepova, I.A.; Chupakhin, O.N. Synthesis and Physicochemical Properties of Nanostructured TiO2 with Enhanced Photocatalytic Activity. Inorg. Mater. 202157, 503–510. https://doi.org/10.1134/S0020168521050022

Manuputty, M. Y., Xu, R., & Kraft, M. (2023). Effects of particle collection in a premixed stagnation flame synthesis of sub-stoichiometric TiO2-x nanoparticles. Chemical Engineering Science265, 118155.

3.     Line 29. It would be useful to end this sentence with the most important references to these impurities in TiO2, which is the subject of this paper. This would increase the interest of this work among readers.

4.     Speaking of using Fluorescence Microscopy, what type of fluorescence is observed?

5.     Please check references 22, 23, where 2 times the year is indicated, but no volume

In general, the manuscript is interesting and can be recommended for publication after constructive reflection on the above comments.

Author Response

Referee report: “MRC-5 Human Lung Fibroblasts Alleviate the Genotoxic Effect of Fe-N Co-Doped Titanium Dioxide Nanoparticles Through  an OGG1/2-Dependent Reparatory Mechanism”

This is a very interesting review article that probably can be recommended for publication, but only after clarifying and detailing some parts of the text.

We would like to thank the reviewer for the kind appreciation of our work. It is noteworthy to mention the manuscript represents an original research article, because contains new data and unpublished previously. 

  1. It is not clear from the second paragraph which structure of the titanium oxide is meant, because TiO2 is known in several crystalline modifications. It is important to note that there are several modification of TiO2 and they give different effects. See, some of them: Tsebriienko, T.; Popov, A.I. Effect of Poly(Titanium Oxide) on the Viscoelastic and Thermophysical Properties of Interpenetrating Polymer Networks. Crystals 2021, 11, 794. https://doi.org/10.3390/cryst11070794

Response: We agree with the reviewer that mentioning the differential toxic effect of crystalline structures of TiO2 NPs would improve our manuscript. We included this information in the Introduction section (lines 38-44), citing the most relevant references:

 [16] Tsebriienko, T.; Popov, A.I. Effect of Poly(Titanium Oxide) on the Viscoelastic and Thermophysical Properties of Interpen-etrating Polymer Networks. Crystals 2021, 11, 794. https://doi.org/10.3390/cryst11070794

[17] Lal, M.; Sharma, P.; Ram, C. Calcination temperature effect on titanium oxide (TiO2) nanoparticles synthesis. Optik 2021, 241, 166934. https://doi.org/10.1016/j.ijleo.2021.166934

[18] Dorosheva, I.B.; Valeeva, A.A.; Rempel, A.A.; Trestsova, M.A.; Utepova, I.A.; Chupakhin, O.N. Synthesis and Physico-chemical Properties of Nanostructured TiO2 with Enhanced Photocatalytic Activity. Inorg. Mater. 2021, 57, 503–510. https://doi.org/10.1134/S0020168521050022

[19] Di Paola, A.; Bellardita, M.; Palmisano, L. Brookite, the Least Known TiO2 Photocatalyst. Catalysts 2013, 3, 36-73. https://doi.org/10.3390/catal3010036

[20] Manuputty, M.Y.; Xu, R.; Kraft, M. Effects of particle collection in a premixed stagnation flame synthesis of sub-stoichiometric TiO2-x nanoparticles. Chem. Eng. Sci. 2023, 265, 118155. https://doi.org/10.1016/j.ces.2022.118155

[21] Ramanavicius, S.; Ramanavicius, A. Insights in the Application of Stoichiometric and Non-Stoichiometric Titanium Oxides for the Design of Sensors for the Determination of Gases and VOCs (TiO2−x and TinO2n−1 vs. TiO2). Sensors 2020, 20, 6833. https://doi.org/10.3390/s20236833

[22] Numano, T.; Xu, J.; Futakuchi, M.; Fukamachi, K.; Alexander, D.B.; Furukawa, F.; Kanno, J.; Hirose, A.; Tsuda, H.; Suzui, M. Comparative Study of Toxic Effects of Anatase and Rutile Type Nanosized Titanium Dioxide Particles in vivo and in vitro. Asian Pac. J. Cancer Prev. 2014, 15, 929-935. https://doi.org/10.7314/APJCP.2014.15.2.929

[23]       Iswarya, V.; Bhuvaneshwari, M.; Chandrasekaran, N.; Mukherjee, A. Individual and binary toxicity of anatase and rutile nanoparticles towards Ceriodaphnia dubia. Aquat. Toxicol. 2016, 178, 209-221. https://doi.org/10.1016/j.aquatox.2016.08.007

  1. What effect does the size of nanoparticles have? How important is stoichiometry?

Dorosheva, I.B.; Valeeva, A.A.; Rempel, A.A.; Trestsova, M.A.; Utepova, I.A.; Chupakhin, O.N. Synthesis and Physicochemical Properties of Nanostructured TiO2 with Enhanced Photocatalytic Activity. Inorg. Mater. 2021, 57, 503–510. https://doi.org/10.1134/S0020168521050022

Manuputty, M. Y., Xu, R., & Kraft, M. (2023). Effects of particle collection in a premixed stagnation flame synthesis of sub-stoichiometric TiO2-x nanoparticles. Chemical Engineering Science, 265, 118155.

Response: Thank you very much for your valuable comment. A brief discussion regarding the effect of size and stoichiometry on the biological activity of TiO2 NPs was added in the Introduction section (lines 45-51) and the most relevant references were cited:

[24] Prokopiuk, V.; Yefimova, S.; Onishchenko, A.; Kapustnik, V.; Myasoedov, V.; Maksimchuk, P.; Butov, D.; Bespalova, I.; Tkachenko, A. Assessing the Cytotoxicity of TiO2−x Nanoparticles with a Different Ti3+(Ti2+)/Ti4+ Ratio. Biol. Trace Elem. Res. 2022, online ahead of print. https://doi.org/10.1007/s12011-022-03403-3

[25] Xiong, S.; George S.; Ji, Z.; Lin, S.; Yu, H.; Damoiseaux, R.; France, B.; Ng, K.W.; Loo, S.C.J. Size of TiO2 nanoparticles in-fluences their phototoxicity: an in vitro investigation. Arch. Toxicol. 2013, 87, 99-109. https://doi.org/10.1007/s00204-012-0912-5

[26] Liu, R.; Yin, L.; Pu, Y.; Liang, G.; Zhang, J.; Su, Y.; Xiao, Z.; Ye, B. Pulmonary toxicity induced by three forms of titanium dioxide nanoparticles via intra-tracheal instillation in rats. Prog. Nat. Sci. 2009, 19, 573-579. https://doi.org/10.1016/j.pnsc.2008.06.020

[27] Hussain, S.; Boland, S.; Baeza-Squiban, A.; Hamel, R.; Thomassen, L.C.J.; Martens, J.A.; Billon-Galland, M.A.; Fleury-Feith, J.; Moisan, F.; Pairon, J.C.; Marano, F. Oxidative stress and proinflammatory effects of carbon black and titanium dioxide nanoparticles: Role of particle surface area and internalized amount. Toxicology 2009, 260, 142-149. https://doi.org/10.1016/j.tox.2009.04.001

[28] Murugadoss, S.; Brassinne, F.; Sebaihi, N.; Petry, J.; Cokic, S.M.; van Landuyt, K.L.; Godderis, L.; Mast, J.; Lison, D.; Hoet, P.H.; van der Brule, S. Agglomeration of titanium dioxide nanoparticles increases toxicological responses in vitro and in vivo. Part. Fibre Toxicol. 2020, 17, 10. https://doi.org/10.1186/s12989-020-00341-7

  1. Line 29. It would be useful to end this sentence with the most important references to these impurities in TiO2, which is the subject of this paper. This would increase the interest of this work among readers.

Response: We would like to thank you for this recommendation, but we think a confusion occurred because of the sentence structure. We did not mean to refer to impurities on TiO2 NPs, but to the most used metals for the production of nanomaterials. We reformulated the phrase (lines 28-29) in order to be clearer.

  1. Speaking of using Fluorescence Microscopy, what type of fluorescence is observed?

Response: The red fluorescent signal in images displaying cathepsin B was emitted by Alexa Fluor 594 following immunofluorescent staining. As for the actin cytoskeleton (green), nuclei (blue), lysosomes analyses (green) and Comet assay (green), the fluorescence comes from biological dyes, namely FITC, DAPI, LysoTracker Green, Vista Green, respectively. This information was detailed in the Materials and Methods section, chapters 4.3 and 4.4 (pages 11-12).

  1. Please check references 22, 23, where 2 times the year is indicated, but no volume.

Response: Thank you for your careful check of the manuscript. These references were published in Hindawi journals that assign the number of volumes with the year in which they were published.

In general, the manuscript is interesting and can be recommended for publication after constructive reflection on the above comments.

We hope that we appropriately addressed the comments and that now the reviewer will be satisfied.

Reviewer 2 Report

This work reports a study on the potential of pure P25 TiO2 nanoparticles (NPs) and Fe(1%)-N co-doped P25 TiO2 NPs to induce cyto- and genotoxic effects in MRC-5 human pulmonary 14 fibroblasts. However, the manuscript does not provide sufficient experiments and evidences to visualize the experiments of the nanoparticles with cells. The manuscript can be accept if additional characterization data could be address. Followings are the reviewer's concerns that need to be addressed:

- Please indicate the rationale on the use of  Fe(1%)-N co-doped P25 TiO2 NPs, why not other percentages such as 3%, 5%. It is better to have upper and lower percentages in the experiments (all biological data).

- Please provide experimental evidences to visualize/quantify the nanoparticles, including undoped and doped ones, by electron microscopes. 

Author Response

This work reports a study on the potential of pure P25 TiO2 nanoparticles (NPs) and Fe(1%)-N co-doped P25 TiO2 NPs to induce cyto- and genotoxic effects in MRC-5 human pulmonary 14 fibroblasts. However, the manuscript does not provide sufficient experiments and evidences to visualize the experiments of the nanoparticles with cells. The manuscript can be accept if additional characterization data could be address. Followings are the reviewer's concerns that need to be addressed:

- Please indicate the rationale on the use of  Fe(1%)-N co-doped P25 TiO2 NPs, why not other percentages such as 3%, 5%. It is better to have upper and lower percentages in the experiments (all biological data).

Response: We tried to reveal in the Introduction section that the biological effect of TiO2 NPs doped with iron has not been well documented, in contrast to the enhanced photocatalytic properties determined by this doping. So far, our work aimed to investigate only if impregnation with iron has an influence on the toxicological response of in vitro cultured cells to the exposure to TiO2 nanoparticles. As we found evidence that modified effects occur, doping of TiO2 nanoparticles with upper and lower percentages of iron would be the subject of a future research investigating the influence of the variation of these percentages on cells’ response.

- Please provide experimental evidences to visualize/quantify the nanoparticles, including undoped and doped ones, by electron microscopes.

Response: The manuscript presented the latest results from our research group on P25 TiO2 NPs and Fe(1%)-N P25 TiO2 NPs. We already published the characterization of the doped and undoped NPs in the previous works (see below the references 83 and 103), including transmission electron microscopy images, so we considered it would not be reasonable to publish these results again. The previous papers that included the characterization of nanoparticles were mentioned in the manuscript (lines 378-382). Also, a brief description of the nanoparticles was provided in the manuscript, in the Materials and Methods section (lines 375-378).

[83] Nica, I.C.; Stan, M.S.; Popa, M.; Chifiriuc, M.C.; Lazar, V.; Pircalabioru, G.G.; Dumitrescu, I.; Ignat, M.; Feder, M.; Tanase, L.C.; Mercioniu, I.; Diamandescu, L.; Dinischiotu, A. Interaction of new-developed TiO2-based photocatalytic nanoparticles with pathogenic microorganisms and human dermal and pulmonary fibroblasts. Int. J. Mol. Sci. 2017, 18, 249. https://doi.org/10.3390/ijms18020249

[103] Diamandescu, L.; Feder, M.; Vasiliu, F.; Tanase, L.; Sobetkii, A.; Dumitrescu, I.; Teodorescu, M.; Popescu, T. Hydrothermal route to (Fe, N) codoped titania photocatalysts with increased visible light activity. Ind. Textila 2017, 68, 303–308. https://doi.org/10.35530/IT.068.04.1438

Round 2

Reviewer 2 Report

I am not satisfied with the response from the authors. I expect the authors should provide the rationale why choosing the doping of 1% Fe and N. Is that just due to the authors have no other choice? Otherwise, readers think any percentage of doping is fine. More importantly, this amount must be quantified. Moreover, with the fast rate of life, readers track the authors previous papers and see what your nanoparticles look like. 

Author Response

Reviewer #2
I am not satisfied with the response from the authors. I expect the authors should provide the rationale why choosing the doping of 1% Fe and N. Is that just due to the authors have no other choice? Otherwise, readers think any percentage of doping is fine. More importantly, this amount must be quantified. Moreover, with the fast rate of life, readers track the authors previous papers and see what your nanoparticles look like.
Response: In the newly revised manuscript the rationale of using 1% FeCl3 to dope TiO2 NPs is provided in the paragraphs between the lines 114-128. Also, we showed the XPS measurements that quantifies the amount of Fe and N on TiO2 NPs (Figure 1, Table 1; lines 129-141) and TEM images of TiO2 NPs (Figure 2; lines 142-149). We hope that we appropriately addressed the comments and that now the reviewer will be satisfied with the additional information. 

Round 3

Reviewer 2 Report

It is good to see the authors trying to make their research outcome be clear to readers by providing the rationale of the selected doped nanoparticles. In fact, for a comprehensive research, the visualization of nanoparticle-cells interaction by microscopes is critically important. However, the authors may not have such facilities.

I would suggest the authors to include the survey spectrum together with their high resolution spectra in Figure 1, thus readers can have the overall picture on the XPS data. 

Author Response

Thank you very much for your suggestion. We included the survey spectrum together with their high resolution spectra in Figure 1 of the revised manuscript, thus readers can now have the overall picture on the XPS data.